# Shattering the Rings: Reproducibility and Vulnerability Analysis of the ZoDiac Watermarking Framework

## Abstract

This paper presents a reproducibility study and robustness evaluation of the paper 'Attack-Resilient Image Watermarking Using Stable Diffusion' by Zhang et al. (2024), which proposes ZoDiac, a Stable Diffusion-based framework for attack-resilient image watermarking. While verifying the original method's core claims achieving >90% watermark detection rate (WDR) against diffusion-based regeneration attacks and across MS-COCO, DiffusionDB, and WikiArt datasets, we identify critical vulnerabilities under adversarial and geometrically asymmetric attack paradigms. Our vulnerability analysis demonstrates that gradient-based adversarial perturbations reduce ZoDiac's WDR, a threat model absent in prior evaluation. We also investigate rotationally asymmetric attacks achieving WDR below 65%. We discover composite attacks combining adversarial noise with other methods reducing WDR to near-zero, exposing vulnerabilities through multi-stage offensive pipelines. Our implementation can be found on Anonymous Github[1] .

## 1 Introduction

The rapid advancement of generative AI has heightened the need for robust image watermarking techniques to verify content authenticity and counter AI-generated forgeries Craver et al. (1998); Tirkel et al. (1994); Cox et al. (2007). Traditional watermarking methods, such as frequency-domain embeddings like the Discrete Cosine Transform (DCT)(Bors & Pitas, 1996; Barni et al., 1998) and Discrete Wavelet Transform (DWT) (Hsu & Wu, 1998), or spatial-domain techniques like Least-Significant-Bit (LSB)(Wolfgang & Delp, 1996) manipulation, were designed to withstand standard distortions such as JPEG compression and Gaussian noise. However, these classical approaches struggle against modern diffusion-based regeneration attacks, which leverage latent-space purification to effectively erase embedded watermarks.

Early neural network-based approaches, including RivaGAN(Zhang et al., 2018) and StegaStamp(Tancik et al., 2020), improved resilience through adversarial training and the use of spatial transformer networks. Yet, because their embeddings operate in the pixel space, they remain vulnerable to pipeline-aware attacks that exploit the iterative denoising process of diffusion models to remove the watermark. This fundamental weakness stems from their operation in pixel space, which makes the embedded marks susceptible to latent-space purification. Recent diffusion-based techniques aim to address this vulnerability. Tree-Ring(Wen et al., 2023) encodes concentric ring patterns into the initial noise vectors of synthetic images, leveraging the deterministic inversion property of diffusion models to recover the watermark from generated outputs. While this method achieves rotational invariance and resists individual attacks by embedding patterns in the Fourier domain, it is only applicable to synthetically generated images, leaving real-world content unprotected. Furthermore, its reliance on isotropic ring patterns creates vulnerabilities to asymmetric transformations, and its static design lacks defenses against composite attacks. Other methods like StableSignature(Fernandez et al., 2023) fine-tune diffusion decoders to embed watermarks but require extensive training on large datasets, making them resource-intensive and impractical for many applications.

ZoDiac(Zhang et al., 2024) addresses these gaps with a novel framework that integrates a pre-trained Stable Diffusion model with DDIM inversion to embed imperceptible watermarks in existing images. The method

---

[1] Link to Anonymous Github

maps an input image into a latent vector via inversion, injecting a ring-shaped watermark into the Fourier domain of this latent space, and reconstructs the image while preserving high fidelity. Unlike pixel-space methods, ZoDiac operates in the latent space, where the iterative denoising process of diffusion models can reinforce the watermark's persistence, making it inherently resistant to purification attacks. ZoDiac also explicitly aligns the injected watermark with its retrieved version in the Fourier space, a mechanism that counters latent-space distortions caused by pixel-space augmentations and distinguishes it from the static, synthetic-only embeddings of Tree-Ring.

We selected the ZoDiac framework for this study primarily because of its innovative approach to embedding watermarks directly into the latent space of a pre-trained Stable Diffusion model. By leveraging DDIM inversion and Fourier-domain ring patterns, ZoDiac claims superior performance over existing SOTA methods, without the computational overhead of fine-tuning the backbone model as compared to previous SOTA methods such as Stable Signature(Fernandez et al., 2023). A key factor in our selection was its purported robustness to geometric attacks, such as rotation and scaling, enabled by its radially symmetric embedding mechanism. While the framework also aims to resist diffusion-based regeneration, its zero-shot flexibility for both real-world and synthetic images makes it a highly practical and sophisticated baseline for investigating the vulnerabilities of latent-space watermarking against adversarial and geometric threats.

Our main contributions are:

- **Verification and Cross-Checking of Original Claims:** We provide a comprehensive reproducibility analysis that validates ZoDiac's performance and core claims under its originally reported conditions.

- **Identification of Geometrically Asymmetric Vulnerabilities:** Our work provides a systematic exposure of sensitive vulnerabilities inherent in ZoDiac. We demonstrate that ZoDiac's reliance on isotropic Fourier-domain ring patterns makes it fragile against asymmetric distortions. While robust to standard Gaussian blur, its performance drops significantly when subjected to directional blurring, which violates its circular symmetry assumptions and reduces the WDR to as low as 64.6%.

- **Identification of Lateral Inversion and Rotational Fragility:** While the original framework was evaluated only for specific, orthogonal rotations and omitted lateral inversions entirely, we find that arbitrary rotational angles and lateral inversions induce phase shifts that misalign the Fourier mask; under these conditions, the WDR fluctuates significantly and frequently collapses.

- **Exposure of Vulnerabilities to Chromatic Distortions:** We identify a critical weakness in the framework's reliance on SSIM for quality control, which is relatively insensitive to hue shifts. Our experiments demonstrate that systematic hue perturbations, color quantization, and sepia filters disrupt watermark alignment in the Fourier domain, significantly impacting WDR.

- **Exposure of Susceptibility to Adversarial Attacks:** We perform gradient-based adversarial perturbations that were absent in prior evaluation. The study reveals that while standalone adversarial noise reduces WDR to approximately 75.5%, composite attacks that combine adversarial noise with geometric distortions destabilize the latent-space structure, rendering the watermark virtually undetectable.

## 2 Related Works

The field of digital image watermarking has evolved through a continuous arms race between embedding techniques and adversarial attacks, progressing from simple spatial-domain manipulations to the sophisticated use of generative models' latent spaces. Early research in the 1990s established a foundational trade-off between three competing objectives: imperceptibility, robustness, and capacity. Initial methods operated in the spatial domain, with techniques like Least Significant Bit (LSB)(Wolfgang & Delp, 1996) substitution offering high capacity at the cost of extreme fragility, rendering them vulnerable to nearly any image processing operation. A significant leap in robustness was achieved with the introduction of spread spectrum watermarking(Cox et al., 1997), which argued that a resilient watermark must be embedded within the most

perceptually significant components of the host signal. This approach treated the watermark as a wideband, noise-like signal embedded across the image's most critical spectral coefficients, providing strong resistance to compression and noise. The limitations of these early methods, particularly their vulnerability to geometric desynchronization, were systematically exposed by research like the StirMark benchmark (Kutter & Petit-colas, 1999), an attack tool that applied minor, random geometric distortions to defeat most contemporary watermarking schemes. This challenge catalyzed a shift toward transform-domain techniques that leverage properties of the Human Visual System (HVS). Methods based on the Discrete Cosine Transform (DCT) offered inherent resilience to JPEG compression(Bors & Pitas, 1996; Barni et al., 1998), while those using the Discrete Wavelet Transform (DWT) (Hsu & Wu, 1998) provided excellent spatio-frequency localization. The Discrete Fourier Transform (DFT) proved particularly effective against the geometric attacks posed by StirMark due to its natural invariance to rotation, scaling, and translation.

The advent of deep learning introduced a paradigm shift, moving from handcrafted features to end-to-end learned watermarking strategies. The predominant encoder-decoder framework trains a neural network to learn an optimal, non-linear transformation for embedding a watermark, guided by a composite loss function that balances imperceptibility and robustness. During training, a non-trainable "attack layer" simulates various distortions, forcing the network to learn a resilient embedding. Architectural innovations, including the use of Convolutional Neural Networks (CNNs) (Tavakoli et al., 2022; Zhu et al., 2018a), Generative Adversarial Networks (GANs) (Ong et al., 2021; Zhang et al., 2018) for enhanced imperceptibility, and attention mechanisms (e.g., DARI-Mark(Zhao et al., 2023b)) for content-aware embedding, have further advanced the state of the art. However, this paradigm also introduced a new attack surface. Deep learning-based systems are vulnerable to adversarial attacks(Choubassi & Moulin, 2005; Comesaña et al., 2006) that craft imperceptible perturbations to cause extraction failure, as well as model-based attacks like fine-tuning or overwriting that can remove or replace the embedded watermark.

Most recently, the proliferation of high-fidelity diffusion models has again reshaped the landscape. These models present a formidable threat through "regeneration attacks", where a watermarked image is denoised and reconstructed, effectively treating the watermark as noise and purging it from the regenerated output. Concurrently, these models offer a powerful new medium for embedding. Instead of modifying pixels, the watermark can be integrated into the generative process itself, making it intrinsic to the image's semantic structure. Tree-Ring Watermarking (Wen et al., 2023), is designed to fingerprint the output of diffusion models during generation. It operates by embedding a predefined pattern into the Fourier space of the initial noise vector before the denoising process begins. The final image is a clean output from the model, yet it carries an indelible, private fingerprint that can be detected only by the model owner, who can invert the diffusion process to inspect the initial noise vector. Addressing the challenge of watermarking existing images, Zodiac (Zhang et al., 2024) leverages a pre-trained stable diffusion model. Its core principle is to inject a watermark not into the image pixels, but into a trainable latent space.

## 2.1 Scope of Reproducibility

In this reproducibility study, we rigorously validate the fundamental claims of the ZoDiac framework while subjecting it to comprehensive stress tests across a range of attack paradigms. Our efforts are primarily directed toward verifying the four central claims outlined in the original work.:

- **Claim 1 :** ZoDiac demonstrates a watermark detection rate (WDR) exceeding 98 and a false positive rate (FPR) below 6.4 across MS-COCO, DiffusionDB, and WikiArt datasets, outperforming state-of-the-art watermarking methods.

- **Claim 2 :** ZoDiac remains resilient to diverse attack categories, including traditional attacks (such as JPEG compression and Gaussian blurring), Stable Diffusion-based regeneration attacks, where most other methods fail and rotational attacks to some extent.

- **Claim 3 :** ZoDiac is highly practical for diverse applications and real world deployment, as it requires no retraining of the Stable Diffusion model, tackling both real-world and synthetic images unlike prior methods like Tree-Ring or Stable Signature.

- **Claim 4 :** ZoDiac achieves imperceptible watermarks with image quality metrics such as SSIM $\geq 0.92$, ensuring minimal visual degradation while maintaining robustness against attacks. ZoDiac ensures a fair tradeoff between watermark detection and maintaining image quality.

# 3 Methodology

## 3.1 Description of ZoDiac Framework

The ZoDiac framework methodology comprises three key components: (1) latent-space vector initialization via DDIM inversion, (2) Fourier-domain watermark embedding for geometric resilience, and (3) adaptive image enhancement to balance detectability and visual fidelity. It employs WDR, PSNR, FPR and SSIM as evaluation criteria (Appendix A).

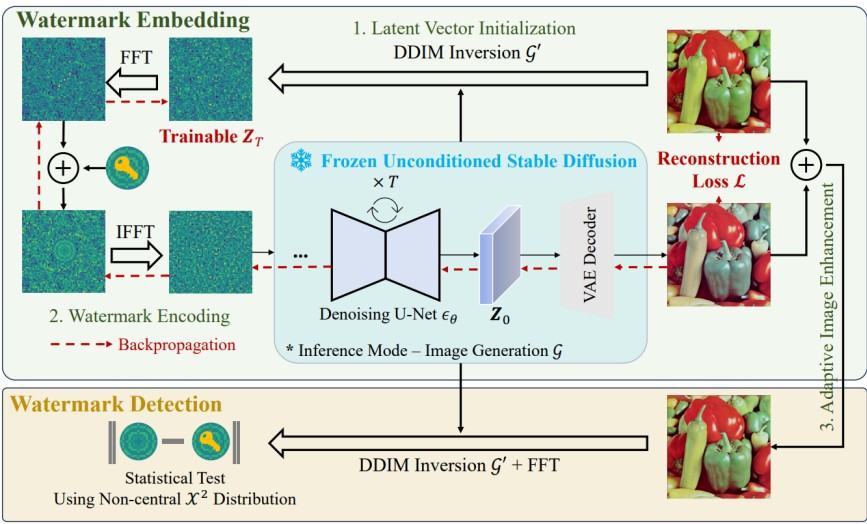

Figure 1: Overview of the ZoDiac framework, illustrating the watermark embedding and detection pipleines. Image taken from the base paper Zhang et al. (2024)

### 3.1.1 Latent Vector Initialization via DDIM Inversion

The process begins by mapping an input image $x_0$ to a latent vector $\mathbf{Z}_T$ using **DDIM inversion**:

$$\mathbf{Z}_T = \mathcal{G}'(x_0), \tag{1}$$

where $\mathcal{G}'$ denotes the inversion of the pre-trained Stable Diffusion model. The inversion adheres to the forward diffusion process:

$$\mathbf{Z}_{t-1} = \sqrt{\bar{\alpha}_{t-1}}\left(\frac{\mathbf{Z}_t - \sqrt{1-\bar{\alpha}_t}\epsilon_\theta(\mathbf{Z}_t, t)}{\sqrt{\bar{\alpha}_t}}\right) + \sqrt{1-\bar{\alpha}_{t-1}}\epsilon_\theta(\mathbf{Z}_t, t), \tag{2}$$

where $\bar{\alpha}_t$ controls the noise schedule and $\epsilon_\theta$ is the pre-trained denoiser. Initializing $\mathbf{Z}_T$ via inversion ensures faster convergence and preserves the structural integrity of the original image during watermark injection.

### 3.1.2 Fourier-Domain Watermark Encoding

ZoDiac injects a ring-shaped watermark $\mathbf{W}$ into the Fourier transform of $\mathbf{Z}_T$ to exploit rotational symmetry and frequency-domain resilience:

**Watermark Generation:** $\mathbf{W}$ is sampled from $\mathcal{CN}(0,1)$ (complex Gaussian distribution), with elements equidistant from the center of latent vector being assigned identical values. A binary mask $\mathbf{M}$ localizes the

watermark to low/mid frequencies:

$$\mathbf{M}_p = \begin{cases} 1 & \text{if } d(p, c) \leq d^* \\ 0 & \text{otherwise} \end{cases}, \tag{3}$$

where $d(p, c)$ is the Euclidean distance from coordinate $p$ to the latent center $c$, and $d^*$ is the mask radius.

**Watermark Injection:** The watermark is applied to the Fourier-transformed latent vector:

$$\mathcal{F}(\mathbf{Z}_T)[ic, :, :] = (1 - \mathbf{M}) \odot \mathcal{F}(\mathbf{Z}_T)[ic, :, :] + \mathbf{M} \odot \mathbf{W}, \tag{4}$$

where $\mathcal{F}(Z_T) \in \mathbb{C}^{ch \times w \times h}$ is the Fourier transform of latent vector, $ic$ is the target watermark injection channel and $\odot$ represents element wise product. We denote the latent vector after watermarking as $\mathbf{Z}_T \oplus \mathbf{W}$.

**Latent Optimization:** The watermarked latent $\mathbf{Z}_T \oplus \mathbf{W}$ is optimized via gradient descent to minimize a multi-term reconstruction loss:

$$\mathcal{L} = \underbrace{\|\hat{x}_0 - x_0\|_2}_{\text{L2}} + \lambda_s \mathcal{L}_{\text{SSIM}} + \lambda_p \mathcal{L}_{\text{Watson-VGG}}, \tag{5}$$

where $\mathcal{L}_{\text{SSIM}}$ is SSIM loss(Zhao et al., 2017) which preserves structural similarity , and $\mathcal{L}_{\text{Watson-VGG}}$ corresponds to the Watson-VGG perceptual loss(Czolbe et al., 2021) enforcing perceptual fidelity using features extracted by pre-trained VGG network.

### 3.1.3 Adaptive Image Enhancement

To preserve visual quality while maintaining watermark robustness, ZoDiac employs an adaptive blending mechanism between the watermarked image $\hat{x}_0$ and original $x_0$ through the parameterized operation:

$$\bar{x}_0 = \hat{x}_0 + \gamma(x_0 - \hat{x}_0), \tag{6}$$

where the blending coefficient $\gamma \in [0, 1]$ is optimized via binary search to satisfy structural fidelity constraints:

$$\min \gamma \quad \text{s.t.} \quad \text{SSIM}(\bar{x}_0, x_0) \geq s^*. \tag{7}$$

This formulation explicitly negotiates the trade-off between imperceptibility and watermark detectability. The dynamic adaptation mechanism enables automatic quality control across diverse image characteristics - a critical improvement over rigid approaches like StegaStamp and RivaGAN. By construction, the blending process preserves high-frequency watermark components while suppressing low-frequency artifacts, ensuring both visual fidelity and attack resilience.

### 3.1.4 Watermark Detection via Statistical Testing

Watermark detection involves performing a null hypothesis test on the presence of $\mathbf{W}$ in binary mask $\mathbf{y}$ of reconstructed image latent $\mathbf{Z}_T$:

1. **DDIM Inversion:** Reconstruct the latent $\mathbf{Z}'_T = \mathcal{G}'(x'_0)$ from the (potentially attacked) image $x'_0$.

2. **Fourier Extraction:** Compute the watermark binary mask $\mathbf{y} = \mathcal{F}(\mathbf{Z}'_T)[-1, :, :]$.

3. **Non-Central Chi-Squared Test:**

   (a) Null hypothesis $H_0 : \mathbf{y} \sim \mathcal{N}(0, \sigma^2 \mathbf{I})$.
   (b) Test statistic: $\eta = \frac{1}{\sigma^2} \sum (\mathbf{M} \odot \mathbf{W} - \mathbf{M} \odot \mathbf{y})^2$. Under $H_0$, $\eta$ follows a non-central chi-squared distribution(Patnaik, 1949)
   (c) Reject $H_0$ if $(1 - p) > p^*$, where $p$ is obtained from the $\chi^2$ CDF with $\sum \mathbf{M}$ degrees of freedom. Here, $(1 - p)$ represents the likelihood of watermark presence and $p^*$ is the set threshold; hence, an image is deemed watermarked if $(1 - p) > p^*$.

## 3.2 Experimental Setup

We obtained the code from the Github[2] repository provided by the original authors and greatly appreciate that their well-structured code was easy to understand and modify. While the original codebase contained the core functionality, though somewhat helpful, it was primarily structured as a demonstration notebook rather than a comprehensive framework for experimental validation. Consequently, one of our key contributions was modifying the original repository to include well-organized and generalizable scripts for all the experiments presented in our paper, as well as for practical applications. All of our code is available here: Anonymous Github[3].

## 3.3 Datasets

**Datasets:** ZoDiac is evaluated across three domains to assess generalizability: **MS-COCO**(Lin et al., 2015) (Real-world photographs, 80,000+ images) to test robustness on natural scenes with complex textures and lighting. A subset of 500 images is randomly sampled from the validation set. **DiffusionDB**(Wang et al., 2023) (AI-generated images, 1.6M+ images) Using a subset of 500 images generated with diverse text prompts. **WikiArt**(Phillips & Mackintosh, 2011) (Artistic works, 250,000+ paintings across 195 styles) validates performance on non-photographic content with unique color palettes and brushstrokes, with a subset of 500 images.

We use an equal number of randomly sampled images from each dataset for all our experiments unless specified otherwise. We use the same datasets as the original paper as we deem the relevance and diversity brought on by these datasets in terms of real and AI-generated images, artistic styles and variety of lighting and textures sufficient for all practical purposes.

## 3.4 Computational Requirements

We evaluated ZoDiac's demands on consumer-grade GPUs for reproducibility. The watermarking pipeline (latent vector initialization and adaptive enhancement) was run on an NVIDIA P100 (16GB VRAM), taking about 295–320 seconds per image (50 denoising steps, 100 optimization iterations). Adversarial attacks (PGD, 50 steps) were executed on an NVIDIA A6000 (48GB VRAM), requiring 820–950 seconds per image due to increased memory needs; these attacks are also feasible on 16GB GPUs at reduced speed. Overall, robustness testing (including DDIM inversion and statistical test) averaged 2 minutes per image on the P100.

| Script | Time (in hours) | Kgs of $CO_2$ |
|---|---|---|
| Basic Watermarking | 13.6 | 2.38 |
| All Attacks (except adv) | 5.3 | 0.92 |
| Adversarial Attacks | 15.8 | 2.77 |

Table 1: GPU usage for a batch of 50 images on two different scripts for 50 denoising steps, 100 training iterations, and 50 steps of PGD on a single P100. Quantity of $CO_2$ estimated using the Machine Learning Impact calculator(Lacoste et al., 2019).

# 4 Reproduction & Verification of Results

We present our results in four subsections, each reproducing a claim made by the base paper and verifying the results.

## 4.1 Verifying Claim 1: Claimed WDR

We evaluated WDR/FPR on 500-image subset from each dataset. We tested with different detection thresholds present our findings in Table 2. As claimed by the original paper, ZoDiac demonstrates superior robust-

---

[2]Link to original paper's GitHub
[3]Link to Anonymous Github

| Detection Threshold | FPR ↓ | Watermark Detection Rate (WDR) ↑ | | | | | | | | | | | |
|---|---|---|---|---|---|---|---|---|---|---|---|---|---|
| | | Pre | Bright. | Cont. | JPEG | G-Noise | G-Blur | BM3D | Bmshj | Cheng | Zhao | Rot. | All |
| MS-COCO | | | | | | | | | | | | | |
| 0.90 | 0.058 | 1.000 | 1.000 | 1.000 | 0.992 | 1.000 | 1.000 | 1.000 | 1.000 | 0.960 | 0.980 | 0.516 | 0.080 |
| 0.95 | 0.014 | 1.000 | 0.996 | 1.000 | 0.988 | 0.998 | 1.000 | 1.000 | 0.920 | 0.958 | 0.974 | 0.316 | 0.080 |
| 0.99 | 0.004 | 0.996 | 0.960 | 0.960 | 0.952 | 0.984 | 0.960 | 0.952 | 0.910 | 0.930 | 0.938 | 0.106 | 0.000 |
| DiffusionDB | | | | | | | | | | | | | |
| 0.90 | 0.052 | 1.000 | 0.998 | 0.998 | 0.988 | 0.978 | 0.984 | 0.960 | 0.980 | 0.990 | 0.956 | 0.530 | 0.070 |
| 0.95 | 0.012 | 1.000 | 0.996 | 0.996 | 0.980 | 0.974 | 0.976 | 0.956 | 0.972 | 0.980 | 0.906 | 0.316 | 0.000 |
| 0.99 | 0.002 | 0.996 | 0.974 | 0.960 | 0.964 | 0.950 | 0.960 | 0.950 | 0.950 | 0.964 | 0.860 | 0.080 | 0.000 |
| WikiArt | | | | | | | | | | | | | |
| 0.90 | 0.058 | 1.000 | 0.980 | 0.980 | 0.980 | 0.980 | 1.000 | 0.980 | 0.976 | 0.960 | 0.980 | 0.428 | 0.040 |
| 0.95 | 0.018 | 1.000 | 0.980 | 0.980 | 0.984 | 0.980 | 1.000 | 0.980 | 0.964 | 0.960 | 0.960 | 0.290 | 0.020 |
| 0.99 | 0.002 | 1.000 | 0.974 | 0.964 | 0.972 | 0.944 | 0.966 | 0.960 | 0.958 | 0.942 | 0.912 | 0.080 | 0.000 |

Table 2: Effects of varying detection thresholds $p^* \in \{0.90, 0.95, 0.99\}$ on watermark detection rate (WDR) and false positive rate (FPR) for all attacks. WDR measured on watermarked images with SSIM threshold $s^* = 0.92$.

ness across diverse datasets (MS-COCO, DiffusionDB, WikiArt) and attack scenarios: adjustments in brightness or contrast,JPEG compression,Image rotation,Gaussian noise,Gaussian blur,BM3D denoising(Dabov et al., 2007),Bmshj(Ballé et al., 2018),Cheng(Cheng et al., 2020),Zhao(Zhao et al., 2023a); achieving >98% Watermark Detection Rate (WDR). Minor discrepancies (<5%) fall within acceptable bounds, reinforcing the original claims' validity. Thus, this claim is verified and there is no further disussion or reason to suggest otherwise.

## 4.2   Verifying Claim 2: Attack Resilience

As demonstrated in our experiments, we successfully reproduced the performance of ZoDiac on all metrics explored in the original paper. However, after identifying limitations inherent to the SSIM metric, we devised novel attack classes designed to exploit these weaknesses. While the authors' claims regarding robustness against the originally evaluated attacks are confirmed as shown in  Table 2, our findings highlight that the general robustness of ZoDiac can be compromised under extended attack scenarios as discussed in 5.

## 4.3   Verifying Claim 3: Deployment Practicality

Watermark injection required 295–320 seconds/image on an NVIDIA P100 GPU (16GB VRAM), aligning with the original paper's reported 255.9s/image on an RTX8000, with minor latency variations attributable to GPU architecture differences. While this per-image latency poses challenges for real-time deployment, ZoDiac's elimination of upfront training costs starkly contrasts with alternatives like Stable Signature, which requires >100 GPU hours to fine-tune the diffusion decoder on a 100K-image dataset.

While ZoDiac's per-image latency exceeds traditional methods like DwtDct(Bloom et al., 1999) (<10s/image), its robustness justifies the trade-off in non-real-time scenarios (e.g., archival systems). Batch processing 100 images parallelized across 4×P100 GPUs reduces effective latency to <2 hours, comparable to Stable Signature's training duration for a single model iteration.

ZoDiac's zero-shot design, provides a viable solution for watermarking existing content without costly retraining. Despite higher per-image latency than non-diffusion methods, its elimination of upfront training (unlike Stable Signature) and dual real/synthetic compatibility (unlike Tree-Ring) make it practical for enterprise-scale deployment. Hence, this claim is also justified.

## 4.4   Verifying Claim 4: Image Quality

Our experiments corroborate ZoDiac's ability to preserve visual fidelity while embedding watermarks, achieving SSIM > 0.91 across all evaluated datasets (MS-COCO, DiffusionDB, WikiArt). These results closely align with the original paper's reported values, with minor variations attributable to stochastic initializa-

tion during latent optimization. The inclusion of SSIM and perceptual losses in ZoDiac's training objective inherently enforces fidelity, ensuring watermarked images remain visually indistinguishable from originals.

While ZoDiac prioritizes imperceptibility, ablation studies reveal a predictable trade-off: stricter SSIM thresholds (e.g., SSIM > 0.95) reduce watermark robustness by nearly 80% WDR under composite attacks. However, the original paper's recommended threshold (SSIM = 0.92) balances this trade-off effectively, as reproduced in our study.

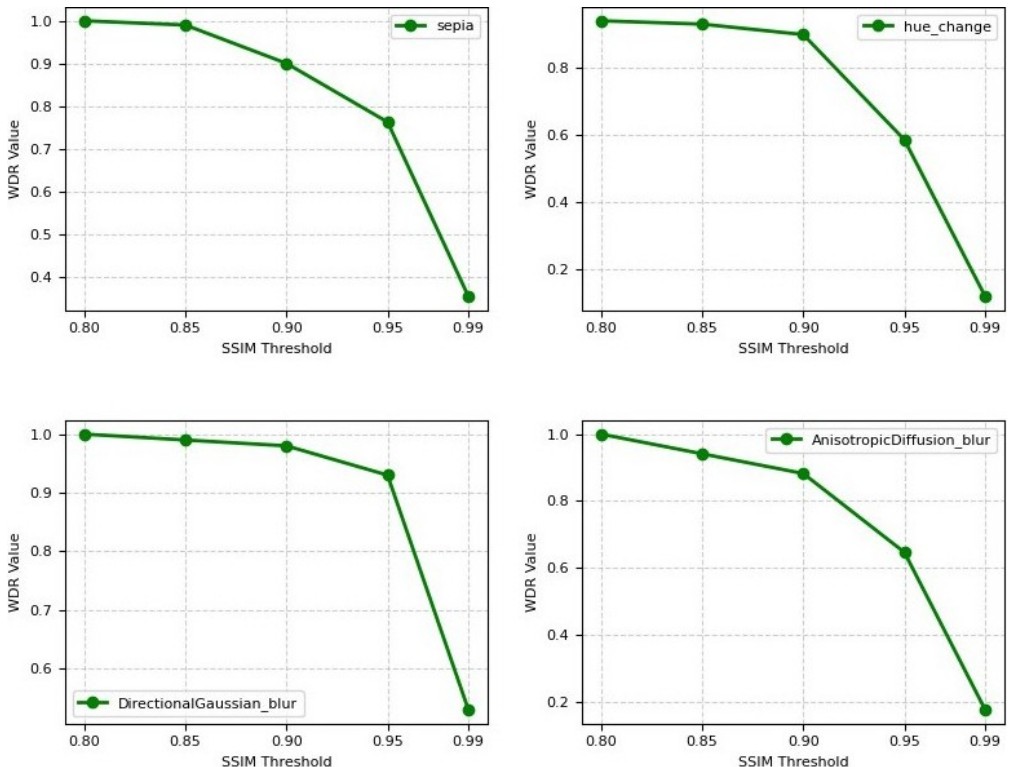

Figure 2: Variation of WDR at $p^* = 0.9$ with different SSIM thresholds for Sepia, Huechange, DirectionalGaussianBlur, AnisotropicDiffusionBur attackers.

The reproduced results validate the claim, confirming ZoDiac's capacity to embed watermarks imperceptibly while preserving image quality. Thus this claim is also verified.

## 5    Vulnerability Analysis

Our extended evaluation reveals that while ZoDiac maintains robustness against its originally tested threat models, the framework is fragile when subjected to more sophisticated attack paradigms, particularly in the hands of a knowledgeable adversary. We found the scheme to be highly susceptible to a range of targeted distortions, including gradient-based adversarial perturbations and geometrically asymmetric manipulations—threat vectors not considered in the initial evaluations. Unlike naturally occurring distortions or standard benchmark attacks, these manipulations are specifically crafted to compromise the watermark's integrity while preserving the perceptual quality of the subject image.

### 5.1    Directional Blurring Attacks

The original paper evaluated robustness against isotropic Gaussian blurs but did not address directional blurring attacks geometrically asymmetric perturbations that exploit rotational dependencies in Fourier-domain watermark embeddings. Our experiments extend ZoDiac's evaluation to directional blurring kernels, revealing latent-space vulnerabilities tied to circular symmetry assumptions of the framework.

ZoDiac's radial Fourier mask $M$ assumes invariance under rotational transformations. Directional blurs violate this assumption, perturbing latent vectors $Z_T$ via:

$$\mathcal{F}(Z_T)_{rot} = R_\theta(\mathcal{F}(Z_T)) \tag{8}$$

where $R_\theta$ denotes rotation by $\theta$. This disrupts the consistency of the DDIM inversion, as transformed latents map to distinct $x_0$ reconstructions. Therefore, we hypothesize that attacking through directional blurring schemes will produce significant change in the WDR of the framework.

The implementation of this experiment considers WDR with baseline blurring (Gaussian Blurring) and two directinal blurring techniques- Anisotropic and Motion Blurring, with the following parameter settings:

| Attack Type | Description | WDR |
|---|---|---|
| Baseline | Isotropic Gaussian blur ($\sigma = 3$) as used in the original study. | 0.994 |
| Motion Blur | Linear kernels (length = 15px) at 30° angles, simulating camera motion. | 0.769 |
| Anisotropic Blur | 45°-aligned Gaussian kernels ($\sigma = 3$) applied along non-radial axes. | 0.646 |

Table 3: Descriptions of Blur Attacks and Baseline

Table 4 displays some of the results obtained while the complete lists is given in B.1.

## 5.2 Changing Color Hue

While ZoDiac demonstrates robustness against diffusion and noise-based attacks, its reliance on SSIM for quality control introduces vulnerabilities to chromatic distortions. Structural Similarity Index (SSIM) emphasizes luminance and structural fidelity but exhibits limited sensitivity to hue shifts a critical gap given real-world attack vectors like selective color grading or adversarial hue perturbations. Our experiments evaluate ZoDiac under systematic hue perturbations, revealing SSIM's failure to capture perceptually significant color distortions that degrade watermark alignment.

Continuing work along these lines , we analyze two more methods - Color Quantization and Sepia filter. Analyzing the attached graphic, Color Quantization and Sepia tone application can be effective at disrupting the ZoDiac watermark. Color Quantization reduces the number of distinct colors in an image, consolidating similar hues into a limited palette. This process compromises high-frequency details and fine gradients. Sepia toning, conversely, is a form of color palette reduction that maps original colors to shades of brown, creating a monochromatic aesthetic. While SSIM aims to capture structural similarity across images, it remains relatively insensitive to broad color palette modifications. For ZoDiac, these attacks can induce misregistrations by distorting relationships between chromatic channels, leading to phase corruptions in the embedded Fourier domain and disrupting the DDIM inversion process during watermark detection. These vulnerabilities highlight that ZoDiac requires additional strategies for watermarking under perceptually relevant color space manipulations.

| | Gaussian | Anisotropic | Sepia Filter | Color Quantization | Hue Change |
|---|---|---|---|---|---|
| **WDR** | 0.994 | 0.646 | 0.902 | 0.807 | 0.773 |

Table 4: Watermark Detection Rate (WDR) for various attack types: blurring kernels and color filters.

## 5.3 Rotational and Lateral Inversion Attacks

In this section, we examine rotation-based and lateral inversion attacks that exploit the geometric dependencies inherent to ZoDiac's Fourier-domain watermark embeddings. ZoDiac embeds watermarks as concentric rings in Fourier space, relying on their radial symmetry to withstand common distortions. However, rotation-based attacks where the image is rotated by arbitrary angles disrupt this symmetry by misaligning the Fourier

mask relative to the original watermark pattern. This misalignment introduces phase shifts that degrade the statistical detection test, resulting in significant drops in the watermark detection rate (WDR).The framework was originally evaluated only for specific 90° rotations, while our analysis, focuses on **arbitrary-angle rotations and lateral inversions**, which induce substantially stronger geometric misalignment in ZoDiac's Fourier-domain watermark, as demonstrated by the marked fluctuations observed with varying rotation angles.(Figure 3)

Similarly, lateral inversion (i.e., horizontal flipping) alters the spatial configuration of the watermark in a manner not anticipated by ZoDiac's detection framework. This inversion modifies the orientation of the latent-space representation, further exacerbating misalignment during DDIM inversion and Fourier-based detection. Both rotation and inversion attacks leverage the absence of inherent geometric invariance in ZoDiac's watermarking scheme, rendering it vulnerable to transformations that change spatial relationships without causing perceptible image quality loss.(Table 6)

We observe that inversion-based transformations cause particularly severe drops in WDR, while preserving perceptual image quality. Although rotation auto-correction has been proposed as a workaround, we find that it exhibits several practical limitations; it increases the FPR by over 10x (0.676 at $p = 0.9$ as compared to 0.062 (Zhang et al., 2024)), forcing the use of high detection thresholds, and it offers no protection against lateral inversions, which are considerably harder to detect and correct in real-world settings.

These observations underscore a key limitation of ZoDiac's design its exclusive reliance on circularly symmetric Fourier embeddings without additional mechanisms for geometric correction. Notably, the combination of rotation and inversion attacks can reduce the WDR to nearly zero, effectively evading watermark detection while preserving image fidelity. Detailed results are displayed in Appendix B.1

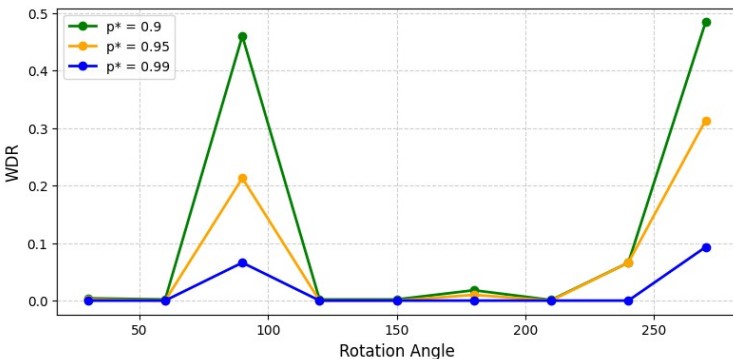

Figure 3: Variation of WDR with change in Rotation Angle at different Detection Threshold ($p^*$)

### 5.4 Adversarial Attack by a knowledgeable adversary

ZoDiac's original robustness claims focus on purification and conventional attacks but omit adversarial perturbations. We extend its threat model to include white-box gradient-based attacks where adversaries perturb the watermarked image to degrade watermark detection. The attacker's objective is to maximize the deviation between the adversarial latent's Fourier components and the original watermark region, constrained by an $\ell_\infty$-norm bound ($\epsilon = 0.05$). This targets the statistical detection mechanism in ZoDiac's Fourier space.

### Attack Methodology

We formulate the adversarial optimization using Projected Gradient Descent (PGD)(Madry et al., 2019) over 50 iterations:

$$Z_T^{adv} = \text{proj}_\epsilon \left( Z_T^{adv} + \alpha \cdot \text{sign} \left( \nabla_{Z_T} L_{adv} \right) \right), \tag{9}$$

where the adversarial loss function is defined as:

$$L_{adv} = \left\| M \odot \left( \mathcal{F}(Z_T^{adv}) - \mathcal{F}(Z_T) \right) \right\|_1. \tag{10}$$

By perturbing the latent vector's Fourier components within this region, the attack disrupts the chi-squared statistical test's assumptions, reducing detection confidence. Under standalone adversarial attacks ($\epsilon = 0.05$), ZoDiac's watermark detection rate (WDR) drops to 75.74%, (Table 5).

Furthermore combining adversarial noise with or geometric attacks amplifies robustness degradation. The synergy arises because adversarial perturbations destabilize the latent vector's Fourier structure, while subsequent attacks exploit residual vulnerabilities. Some of these results are tabulated in Table 5 while a complete list is provided in Appendix D.

**Infeasibility of Adversarial Training**

Adversarial training fine-tuning ZoDiac on adversarially perturbed latents is computationally prohibitive. Each PGD iteration requires 820–950 seconds/image on an NVIDIA A6000 GPU. A standard 50-iteration training protocol would demand over 14 hours per image, translating to more than 21,000 GPU hours for a 1,500-image evaluation set. This stems from backpropagation through the full denoising process during latent optimization, which cannot be parallelized due to DDIM's sequential nature.

The computational estimation mentioned above is an extrapolation derived from the current computational time. This figure does not capture the exact training time, rather it represents an estimate of the time needed to generate adversarial noise for the entire dataset. The computational demand is so high that proper empirical validation is impractical. Alternative approaches, such as pre-generating samples or using faster methods (Wong et al. (2020); Huang et al. (2023) which typically speed up the training by about 5 times, reducing the estimate to roughly 4,200 GPU hours), can help reduce this burden. Though, even with these optimizations, the baseline cost remains impractically high. The effectiveness of adversarial training as a defense remains a critical area for future work, though it currently falls outside our computational budget. Although defenses like randomized thresholds or Fourier-space noise injection could mitigate attacks without any additional training, they might also inadvertently damage watermark detection.

**Implications**

ZoDiac's latent-space watermarking, while robust to purification, is vulnerable to coordinated adversarial-geometric attacks. Future work should explore lightweight defenses, such as Fourier-domain noise augmentation during watermark injection, to disrupt gradient-based exploits without retraining.

| Attack Type | WDR |
|---|---|
| Adversarial ($\epsilon = 0.05$) | 0.755 |
| Adversarial + DiffAttacker60 | 0.491 |
| Adversarial + Rotation(180) | 0.018 |

Table 5: WDR under standalone adversarial attack and composition of adversarial attack with other attacks.

# 6    Conclusion

Our reproducibility study validates ZoDiac's core premise of latent-space watermarking via a pre-trained Stable Diffusion model, achieving robust detection rates (WDR > 98%) and high perceptual quality (SSIM> 0.91) under conventional attack scenarios. However, our extended vulnerability analysis reveals critical weaknesses in the framework's reliance on isotropic Fourier-domain patterns. Specifically, we identify that the system is highly sensitive to geometrically asymmetric attacks, such as directional blurring, and lacks inherent invariance to arbitrary rotational angles and lateral inversions, which cause detection rates to fluctuate significantly or collapse entirely.

We further identify a weakness in the framework's reliance on SSIM for quality control, as the metric exhibits limited sensitivity to hue shifts. Our experiments demonstrate that systematic chromatic distortions, including hue perturbations, color quantization, and sepia filters effectively disrupt watermark alignment in the Fourier domain, with hue changes alone reducing the WDR to approximately 77.3%.

Furthermore, we expose a significant susceptibility to white-box adversarial perturbations. Our results demonstrate a dangerous synergy in multi-stage composite attacks, where adversarial noise combined with geometric transformations renders the watermark virtually undetectable. These results underscore the need for further research into rotation-invariant embeddings and efficient adversarial defenses that can harmonize imperceptibility, robustness, and computational practicality.

## 7 Future Directions

Our analysis is subject to certain limitations and challenges that pave the way for future research. One primary limitation is the nature of our adversarial attack analysis, which assumed a white-box setting. While our focus was on demonstrating the feasibility of incorporating adversarial attacks into the evaluation framework rather than conducting an exhaustive study of partial-knowledge scenarios, we acknowledge that higher PGD budgets can introduce visible artifacts in the generated images. The adaptation of these attacks for settings with incomplete information remains a potential area for exploration, outside our current scope.

Future research efforts should focus on implementing defense modules to counter rotational and lateral-inversion-based attacks that use a pre-calculated orientation anchor to realign the image before the statistical detection test is performed. This approach can help recover WDR while maintaining a low false positive rate (FPR) through comprehensive training, fine-tuning, and parameter selection. Additionally, instead of relying on a static circular mask, the framework could maybe utilize a secret-key-driven, sparse distribution of frequency coordinates, which might disrupt gradient-based adversarial exploits by forcing the attacker to guess the target injection sites. Finally, researchers may consider a more effective reconstruction loss term alongside the extracted watermark within the loss function. Integrating color-aware loss functions, such as those based on CIELAB Delta-E or learned perceptual metrics that prioritize hue consistency, could better safeguard watermark alignment against the phase corruptions induced by color grading.

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

## A    Evaluation Metrics

To comprehensively assess ZoDiac's performance, we use four quantitative metrics spanning detection robustness, image quality and attack resilience. These metrics align with established benchmarks in the field.

**Detection Robustness:**   The **Watermark Detection Rate (WDR)** is calculated as $\text{WDR} = \frac{\text{TP}}{\text{TP + FN}}$, measuring the proportion of watermarked images correctly identified under attack. The **False Positive Rate (FPR)** is given by $\text{FPR} = \frac{\text{FP}}{\text{FP + TN}}$.

**Image Quality Preservation:**   This is measured by evaluating

- **Peak Signal-to-Noise Ratio (PSNR):** Defined as

$$\text{PSNR}(x, \bar{x}) = -10 \log_{10}(\text{MSE}(x, \bar{x}))$$

- **Structural Similarity Index (SSIM):** (Zhao et al., 2017) Enforced as SSIM $\geq 0.92$ through adaptive blending.

# B  Extended Attack Results

## B.1  Comprehensive Attack Performance and Configurations

Here is the list of configurations of all the attacks we used in this paper. Attacks missing from this list either did not have any hyperparameters or the settings were self explanatory.

**Hyperparameter Selections**

- **DiffWM Attack**: Noise step $= 60$

- **Cheng20-Anchor Compression**: Quality $= 3$

- **Bmshj18-Factorized Compression**: Quality $= 3$

- **JPEG Compression**: Quality $= 1$

- **Rotation Attack**: Degree $= 30$

- **Brightness Attack**: Brightness $= 0.5$

- **Contrast Attack**: Contrast $= 0.5$

- **Vibrancy Attack**: Vibrancy $= 1.25$

- **Gaussian Noise Attack**: Standard Deviation $= 0.05$

- **Gaussian Blur Attack**: Kernel Size $= 5$, $\sigma = 1$

- **Anisotropic Diffusion Blur Attack**: Num Iter $= 15$, $\delta_t = 0.14$, $\kappa = 50$

- **Directional Gaussian Blur Attack**: Kernel Size $= 15$, $\sigma = 5$, Angle $= 45$

- **Sharpening Attack**: Factor $= 2.0$

- **Salt and Pepper Noise Attack**: Amount $= 0.1$

- **Hue Change Attack**: Factor $= 0.1$

- **Elastic Deformation Attack**: $\alpha = 1000$, $\sigma = 50$

- **RGB to HSV Attack**: H-Shift $= 0.1$, S-Scale $= 1.2$, V-Scale $= 1.1$

- **Color Balance Attack**: R-Scale $= 1.2$, G-Scale $= 1.0$, B-Scale $= 0.8$

- **Gamma Correction Attack**: $\gamma = 1.5$

- **Log Transform Attack**: c $= 1$

- **Color Jitter Attack**: Brightness $= 0.2$, Contrast $= 0.2$, Saturation $= 0.2$, Hue $= 0.1$

- **Color Quantization Attack**: Number of Colors $= 32$

- **Posterization Attack**: Levels $= 4$

**Results:**

Note: Adversarial attack configurations discussed separately in Section D. Composite attacks ('all', 'all_norot') use parameter unions from above.

| Attacker | 30° | 60° | 90° | 120° | 150° | 180° | 210° | 240° | 270° | Lat. Rot. | Lat. Inversion |
|---|---|---|---|---|---|---|---|---|---|---|---|
| WDR | 0.004 | 0.002 | 0.460 | 0.002 | 0.002 | 0.018 | 0.001 | 0.002 | 0.484 | 0.007 | 0.837 |

Table 6: Variation of WDR with the rotation angle and lateral inversion or a combination (Lat. Rot. means Lateral Inversion + Rotation by 180°)

| Attacker Name | WDR at 0.9 | WDR at 0.95 | WDR at 0.99 |
|---|---|---|---|
| DiffWM Attacker | 0.933 | 0.873 | 0.727 |
| Black and White Attack | 0.913 | 0.853 | 0.787 |
| Lateral Inversion Attack | 0.960 | 0.940 | 0.853 |
| Sharpening Attack | 0.987 | 0.987 | 0.960 |
| Salt and Pepper Noise Attack | 0.833 | 0.673 | 0.427 |
| Hue Change Attack (0.3) | 0.773 | 0.673 | 0.500 |
| Elastic Deformation Attack | 0.967 | 0.967 | 0.927 |
| RGB to HSV Attack | 0.980 | 0.953 | 0.920 |
| Color Balance Attack | 0.993 | 0.973 | 0.940 |
| Gamma Correction Attack | 0.987 | 0.980 | 0.947 |
| Histogram Equalization Attack | 0.993 | 0.973 | 0.933 |
| Log Transform Attack | 1.000 | 1.000 | 0.940 |
| Color Jitter Attack | 0.960 | 0.933 | 0.887 |
| Color Quantization Attack | 0.807 | 0.726 | 0.600 |
| Sepia Attack | 0.900 | 0.840 | 0.773 |
| Posterization Attack | 0.960 | 0.940 | 0.907 |
| Directional Gaussian Blur Attack | 0.947 | 0.933 | 0.853 |

Table 7: WDR of all the attacks other than the Rotation based attacks at different $p^*$ values.

## B.2  Hyperparameter Sensitivity Analysis for Chromatic Distortion Attacks

We extend the original paper's brightness ($\delta_{bright}$) and contrast ($\gamma$) parameter analysis with additional hue variation studies ($\Delta h$). The attack transformations are:

- **Brightness**: $\mathcal{I}' = \text{clip}(\mathcal{I} \cdot \delta_{bright})$ for $\delta_{bright} \in \{0.2, 0.4, 0.5, 0.6, 0.8, 1.25, 1.5\}$

- **Contrast**: $\mathcal{I}' = \mu + \gamma(\mathcal{I} - \mu)$ for $\gamma \in \{0.2, 0.4, 0.5, 0.6, 0.8, 1.25, 1.5\}$

- **Hue(Our Extension)**: $\mathcal{I}'_{HSV} = \mathcal{I}_{HSV} + \Delta h$ for $\Delta h \in \{0.1, 0.3, 0.5, 0.7\}$ in HSV space

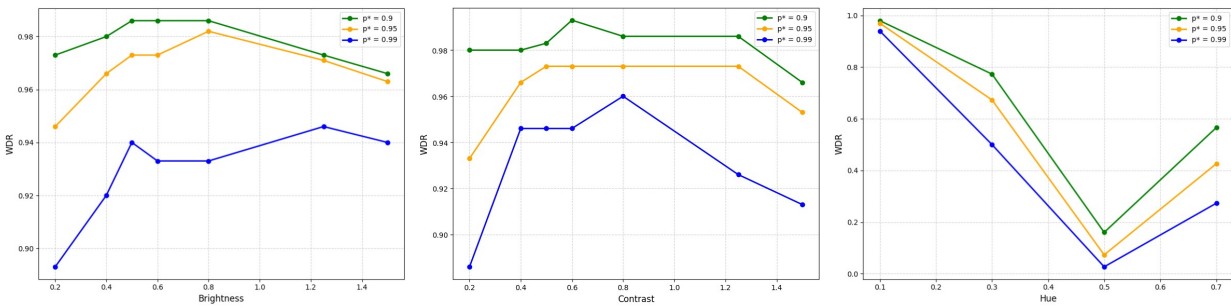

Figure 4: Variation of attack success rates for brightness, contrast (original paper's parameters) and hue variations (our extension) with change in their respective parameters ($\delta_{bright}, \gamma, \Delta h$).

## C    Impact of Reconstruction Loss Term

**Augmenting the Loss Function**

The original ZoDiac framework employs the reconstruction loss:

$$L = \|\hat{x}_0 - x_0\|^2 + \lambda_s L_{\text{SSIM}} + \lambda_p L_{\text{Watson-VGG}} \tag{11}$$

to preserve image fidelity while optimizing latent vectors. While effective in basic reconstruction scenarios, this formulation lacks explicit constraints on watermark alignment between injected $\mathbf{W}$ and reconstructed $\hat{\mathbf{W}}$ patterns, potentially allowing latent-space misregistrations under complex attack scenarios.

Following established principles in the watermarking domain, we integrate direct watermark fidelity into the loss function by incorporating an $L_1$ distance term. This methodology aligns with conventional watermarking techniques that leverage a combination of $L_1$ and $L_2$ constraints to concurrently uphold perceptual quality and enhance watermark robustness(Zhu et al., 2018b; Volpi & Tuia, 2018; Wan et al., 2019; Zhang et al., 2019; Li et al., 2019).

**Mechanistic Analysis**

The $L_1$ term directly minimizes the Manhattan distance between $W$ and $\hat{W}$ in the complex Fourier domain:

$$\|W - \hat{W}\|_1 = \sum_p \left| \text{Re}(W_p - \hat{W}_p) \right| + \left| \text{Im}(W_p - \hat{W}_p) \right| \tag{12}$$

This enforces phase consistency in concentric rings, reducing misalignment under attack-induced perturbations. Our augmented loss thus becomes:

$$L = \underbrace{\|\hat{x}_0 - x_0\|^2}_{L_2} + \lambda_s L_{\text{SSIM}} + \lambda_p L_{\text{Watson-VGG}} + \lambda_{L1} \|W - \hat{W}\|_1 \tag{13}$$

where $\lambda_{L1} = 0.1$ balances watermark alignment and visual fidelity.

Empirical evaluation demonstrates that this approach introduces complex trade-offs between different attack resilience mechanisms, highlighting the fundamental challenge in watermarking systems that simultaneously optimize for multiple conflicting objectives, i.e., imperceptibility, robustness, and capacity. The $L_1$ constraint notably increased resilience to most attack methods, particularly enhancing detection rates for color manipulation attacks (Sepia filters and Color Quantization showing 5.0% and 9.3% improvements respectively). However, some minor performance variations were observed across different attack vectors, with some showing practically no change in WDR, and some infact achieving better attack success.

| Attacker | 30° | 60° | 90° | 120° | 150° | 180° | 210° | 240° | 270° | Lat. Rot. | Lat. Inversion |
|----------|-----|-----|-----|------|------|------|------|------|------|-----------|----------------|
| Original | 0.004 | 0.002 | 0.460 | 0.002 | 0.002 | 0.018 | 0.001 | 0.002 | 0.484 | 0.007 | 0.837 |
| Modified | 0.004 | 0.002 | 0.550 | 0.002 | 0.000 | 0.020 | 0.000 | 0.001 | 0.500 | 0.006 | 0.950 |

Table 8: Rotation-Based Attacks with original and modified loss for comparison

Although the introduction of Reconstruction Loss did mitigate some of the attacks, it also diminished performance in some other attacks. Overall, this loss function can be improved further by calibrating the weights assigned to each loss better or even introducing a new loss term other than $L_1$ loss we have shown.

| Excluded Attacker Names | Original | Modified |
|---|---|---|
| Brightness 0.5 | 0.987 | 0.975 |
| Contrast 0.5 | 0.973 | 0.975 |
| JPEG Compression | 0.687 | 0.775 |
| Gaussian Noise Attack | 0.993 | 0.975 |
| Gaussian Blur Attack | 0.993 | 0.975 |
| BM3D Attack | 0.993 | 1.000 |
| BMSHJ2018-Factorized Compression | 0.960 | 0.950 |
| Cheng2020-Anchor Compression | 0.973 | 0.975 |

Table 9: Original Attacks with original and modified loss for comparison

| Attacker Name | Original | Modified |
|---|---|---|
| DiffWM Attacker | 0.933 | 0.900 |
| Black and White Attack | 0.913 | 0.975 |
| Lateral Inversion Attack | 0.960 | 0.950 |
| Sharpening Attack | 0.987 | 0.975 |
| Salt and Pepper Noise Attack | 0.833 | 0.775 |
| Hue Change Attack (0.3) | 0.773 | 0.775 |
| Elastic Deformation Attack | 0.967 | 0.950 |
| RGB to HSV Attack | 0.980 | 0.950 |
| Color Balance Attack | 0.993 | 1.000 |
| Gamma Correction Attack | 0.987 | 0.975 |
| Histogram Equalization Attack | 0.993 | 1.000 |
| Log Transform Attack | 1.000 | 1.000 |
| Color Jitter Attack | 0.960 | 0.950 |
| Color Quantization Attack | 0.807 | 0.900 |
| Sepia Attack | 0.900 | 0.950 |
| Posterization Attack | 0.960 | 0.975 |
| Anisotropic Diffusion Blur Attack | 0.646 | 0.675 |
| Directional Gaussian Blur Attack | 0.947 | 0.950 |

Table 10: Attacks proposed in our work with original and modified loss for comparison

# D   Adversarial Attack Analysis

## D.1   Composite Attack Results

Here we show the results for combinations of most successful attacks with adversarial attacks. All attacks are performed in standard configurations mentioned in Section B.1, an adversarial attack is carried out with a budget of 0.05 and 50 PGD steps.

## D.2   Attack Budget and Step Count Analysis

We analyze the interaction between perturbation budgets $\epsilon \in \{0.05, 0.1\}$ and PGD step counts $k \in \{5, 10, 20, 50\}$.

Although WDR did go down for higher budgets as expected , due to variations specifically in fourier space, the image quality suffers due to artifaction in the images. Thus the budget should be regulated closely as per the goals of the adversary. The variation of WDR in composite attacks is also almost in proportion to changes in the base case.

| Attack Name | WDR Score |
|---|---|
| Base | 0.7547 |
| diff_attacker_60 | 0.4906 |
| jpeg_attacker_50 | 0.7736 |
| brightness_0.5 | 0.7736 |
| Motion_blur | 0.5094 |
| contrast_0.5 | 0.7358 |
| vibrancy_1.25 | 0.7925 |
| black_white | 0.3962 |
| lateral_inversion | 0.6038 |
| Gaussian_blur | 0.7925 |
| AnisotropicDiffusion_blur | 0.3774 |
| DirectionalGaussian_blur | 0.4906 |
| sharpening | 0.7358 |
| salt_pepper_noise | 0.4151 |
| hue_change_0.5 | 0.0755 |
| posterization | 0.6981 |
| sepia | 0.4151 |
| rotate_90 | 0.0943 |
| rotate_180 | 0.0189 |
| rotate_270 | 0.1887 |
| lateral_rotate | 0.0189 |
| bm3d | 0.7736 |
| all | 0.0566 |

Table 11: WDR Scores for Various Attack Composition with Adversarial Attack

| Budget | Number of Steps | | | |
|---|---|---|---|---|
| | 5 | 10 | 20 | 50 |
| **0.05** | 0.8302 | 0.8278 | 0.8112 | 0.7546 |
| **0.1** | 0.6102 | 0.6032 | 0.5894 | 0.5206 |

Table 12: WDR with different combinations of perturbation budgets and number of PGD optimization steps.

## E    Image Quality Analysis Under Attack-Induced Distortions

A common concern when evaluating robustness metrics such as Watermark Detection Rate (WDR) under aggressive attack pipelines is whether observed performance degradation is merely an artifact of severe image quality loss, rather than a genuine failure of the watermarking mechanism. To explicitly rule out this confounding factor, we conduct a complementary analysis of image fidelity under all attacks introduced in our extended evaluation, using Peak Signal-to-Noise Ratio (PSNR) as a quantitative measure.

Across the complete set of attacks considered in this workincluding geometric distortions, chromatic manipulations, filtering-based operations, and noise injectionsthe average PSNR values lie within a bounded and reasonable range. Specifically, PSNR values span from 15.16 to 38.25, with the majority of attacks clustering well above 18 . This range is fully comparable to and in several instances surpasses the PSNR values reported for standard attacks evaluated in the ZoDiac paper (Zhang et al., 2024), including the Gaussian noise attack with a PSNR of 27.27, and the brightness and contrast attacks, which reach PSNR values as low as 11.41 and 14.54, respectively.

Notably, several attacks that induce substantial WDR degradation maintain PSNR values close to or above this baseline. This observation has two implications. First, the attacks introduced in our extended evaluation do not rely on excessive or visually destructive perturbations; instead, they operate within a distortion regime that remains broadly acceptable by conventional image quality standards. Second, and more critically, we find no consistent or monotonic relationship between PSNR and WDR reduction. Attacks with relatively

high PSNR (e.g., color quantization at 27.84) can still induce meaningful drops in watermark detection, while some lower-PSNR attacks exhibit comparatively modest impact (e.g., log transform at 10.812). This decoupling reinforces the conclusion that WDR failures arise from structural and algorithmic vulnerabilities in the watermarking framework, rather than from trivial signal destruction. For completeness, Table E reports the average PSNR for clean images and for attacks evaluated in this paper.

| Attack | Average PSNR (dB) |
|---|---|
| Black_White | 16.803 |
| Gaussian_Noise | 27.276 |
| Gaussian_Blur | 31.097 |
| Motion_Blur | 19.164 |
| OutofFocus_Blur | 20.819 |
| Radial_Blur | 38.258 |
| Zoom_Blur | 15.666 |
| Atmospheric_Blur | 20.969 |
| PSF_Blur | 21.460 |
| Bilateral_Blur | 29.579 |
| Iterative_Blur | 22.624 |
| AnisotropicDiffusion_Blur | 18.598 |
| DirectionalGaussian_Blur | 20.308 |
| Sharpening | 28.215 |
| Salt_Pepper_Noise | 15.164 |
| Elastic_Deformation | 19.493 |
| RGBtoHSV | 18.509 |
| Color_Balance | 22.481 |
| Gamma | 18.967 |
| Histogram_Equalization | 21.385 |
| Color_Jitter | 20.575 |
| Color_Quantization | 27.841 |
| Sepia | 16.913 |
| Log_Transform | 10.812 |
| Posterization | 27.074 |
| Adversarial | 19.671 |

Table 13: Average PSNR (dB) of images under different attacks.

