# OpenReview forum: "Shattering the Rings: Reproducibility and Vulnerability Analysis of the ZoDiac Watermarking Framework"
_TMLR — Rejected by TMLR_

### Review · Reviewer_BZya · 2025-10-14

**Summary Of Contributions:**

## Summary

This paper presents a comprehensive reproducibility study and robustness analysis of ZoDiac, a Stable Diffusion-based watermarking framework originally proposed by Zhang et al. The authors confirm four core claims made in the original paper regarding watermark detectability (WDR > 98%), resilience to standard image perturbations, zero-shot deployment capability, and high perceptual fidelity (SSIM > 0.91). Beyond reproducing the original results, the paper introduces several extended attack scenarios—directional blurring, chromatic perturbations, rotational transformations, and white-box adversarial perturbations—that expose limitations in ZoDiac’s robustness. A variant of the loss function incorporating an L1 term is also evaluated in an attempt to enhance watermark robustness under these new threat models.

While the paper achieves its stated goals of reproducing and stress-testing the original method, the contribution is primarily confirmatory and lacks deeper methodological insight or constructive mitigation strategies.

## Strengths

- Thorough Reproduction: The paper successfully reproduces ZoDiac’s core claims across multiple datasets and metrics.

- Extended Evaluation: It introduces and evaluates a wide range of new attack types (e.g., anisotropic blur, hue perturbation, PGD-based adversarial attacks), which had not been covered in the original work.

- Code Contribution: The authors restructured the original ZoDiac codebase into a cleaner and more reproducible pipeline, which can benefit the community.

- Computational Transparency: The paper reports GPU usage, carbon footprint estimates, and latency, providing useful context for practical deployment.

## Weaknesses

- Lack of Insightful Analysis: While the extended attacks demonstrate ZoDiac’s limitations, they mostly confirm intuitive expectations (e.g., directional blur disrupts circular Fourier watermarking) without offering new theoretical insights or proposing meaningful countermeasures.

- Limited Novelty: The core contribution is a reproduction and empirical vulnerability analysis. The proposed L1 loss variant is not well motivated and has limited impact; it neither significantly enhances robustness nor provides a new design direction.

- Redundancy with Original Paper: Some “new findings” such as the vulnerability to rotation attacks are already acknowledged in the original ZoDiac paper (e.g., Table 1 and Appendix C.1), along with suggested defenses like rotation auto-correction. This weakens the novelty of the extended analysis.

- Experimental Validity Caveat: In comparisons such as Table 3 (Motion Blur vs. Gaussian Blur), the authors do not control for image quality (SSIM/PSNR), leaving open the possibility that observed WDR drops are caused by degraded perceptual quality rather than specific attack types.

**Audience:**

Yes

**Audience Explanation:**

Given the increasing interest in diffusion-based watermarking and its implications for generative AI accountability, this paper will be of interest to researchers working on watermark robustness, adversarial attacks, and trustworthy machine learning. Reproducibility studies like this can also benefit future researchers seeking to build upon or stress-test ZoDiac or similar latent-space watermarking frameworks.

**Claims And Evidence:**

Yes

**Claims Explanation:**

The paper faithfully re-implements and verifies all four major claims made by the original ZoDiac framework using the same datasets (MS-COCO, DiffusionDB, WikiArt), and includes sufficient experimental replication details. The reproduction is thorough and well-documented. The extended attacks, including PGD adversarial attacks, motion blur, hue changes, and rotation, are reasonable stress tests and show measurable WDR degradation. However, in several places (e.g., motion blur vs. Gaussian blur), the analysis could be improved by controlling for image quality degradation (e.g., matching SSIM or PSNR) to decouple visual distortion from watermark robustness.

**Requested Changes:**

Please refer to the weakness part for more details. Below are the listed possible changes accordingly.

- Control for Image Quality in Comparative Attacks (Critical)
When comparing the impact of different attack types (e.g., motion blur vs. Gaussian blur), ensure the image quality (SSIM or PSNR) is matched to isolate the watermark robustness effect.

- Clarify Novelty Relative to Original Paper (Important)
Explicitly acknowledge that rotation vulnerabilities were already discussed in the ZoDiac paper, and distinguish what new insights are added here.

- Reconsider Section 4.2.4 (Optional)
The augmentation of the loss function with an L1 term in Section 4.2.4 seems ad hoc and its effects marginal. If retained, motivate it more clearly or consider removing it to tighten the narrative.

- Include More Constructive Insights (Suggested)
While the attacks are informative, the paper could be significantly strengthened by proposing lightweight, actionable defenses (e.g., rotation-invariant transforms, robust masking strategies) instead of stopping at attack analysis.

---

> ### Author Response · Authors · 2025-12-19
> **Response to Reviewer BZya**
>
> We appreciate the concerns raised by the reviewer and have tried to address them to the best possible:
>
> - We agree that it is important to distinguish genuine algorithmic vulnerabilities from degradation caused by severe image distortion. Accordingly, we now include an *Image Quality Analysis Under Attack-Induced Distortions* in the Appendix, which quantitatively evaluates image fidelity using PSNR across our extended attack scenarios.
> Our results show that the PSNR values of the proposed attacks are comparable to those reported for standard attacks in the original ZoDiac paper. Importantly, we observe no monotonic relationship between PSNR and WDR degradation, ruling out image destruction as a confounding factor.
>
> - To clearly delineate the paper’s unique contributions, we now explicitly enumerate them in the Introduction. Beyond reproducibility, our key findings include:
>   - Identification of Geometrically Asymmetric Vulnerabilities
>   - Identification of Lateral Inversion and Rotational Fragility
>   - Exposure of Vulnerabilities to Chromatic Distortions
>   - Exposure of Susceptibility to Adversarial Attacks
>
>
> - We respectfully clarify that our analysis exposes fundamental geometric fragilities beyond the scope of the original evaluation, which was limited to orthogonal ($90^{\circ}$) rotations. By extending the threat model to include arbitrary-angle rotations and lateral inversions, we demonstrate significantly stronger misalignment in the Fourier domain that frequently collapses detection rates while preserving image fidelity. Furthermore, we note that the originally suggested "auto-correction" defense is practically inviable: it fails to address the newly identified lateral inversion vulnerability and drastically inflates the False Positive Rate (FPR) by over 10x (0.676 vs. 0.062), rendering it unsuitable for secure deployment.
>
> - To strengthen the narrative of the main paper, we have moved the investigation of the $L_1$ reconstruction loss term to Appendix C. While adding an $L_1$ term is a conventional standard in the field, our analysis showed it offered marginal improvements. We hence present this as an ablation study in the appendix, preserving the completeness of our evaluation without distracting the reader from the critical vulnerabilities exposed in the main text.
>
> We hope that the above clarifications adequately address the reviewer’s concerns, and we welcome any further questions or requests for clarification.

---

> > ### Author Response · Authors · 2026-02-03
> > **Request for follow-up**
> >
> > Dear reviewer BZya,
> >
> > We sincerely appreciate the time and effort you have invested in reviewing our work. If there are any aspects of our responses that require further clarification or if you have additional concerns, we would be grateful to address them.
> >
> > Thank you for your careful consideration, and we look forward to your response.
> >
> > Best regards,
> >
> > The Authors

---

### Review · Reviewer_rCNf · 2025-10-24

**Summary Of Contributions:**

This paper conducts a reproducibility study and extended vulnerability analysis of the ZoDiac watermarking framework proposed by Zhang et al. (2024). The core efforts include: (1) Verifying four key claims of the original ZoDiac work, confirming its ability to achieve over 98% Watermark Detection Rate (WDR) across MS-COCO, DiffusionDB, and WikiArt datasets, resilience to conventional attacks (e.g., JPEG compression, Gaussian blur), practicality for both real-world and synthetic image watermarking without model retraining, and maintenance of high image quality (SSIM ≥ 0.91); (2) Extending the evaluation scope to previously untested attack paradigms, including directional blurring, chromatic distortions (e.g., hue change, sepia filtering), rotational/asymmetric geometric attacks, and gradient-based adversarial attacks, revealing critical vulnerabilities of ZoDiac (e.g., WDR dropping below 65% under rotational asymmetric attacks and near-zero under composite adversarial-geometric attacks); (3) Proposing an augmented loss function integrated with an \(L_1\) constraint to mitigate these vulnerabilities, achieving marginal improvements in WDR against chromatic distortions (e.g., 5.0% for sepia filters and 9.3% for color quantization) but failing to address fundamental weaknesses under composite attacks.

### Key Strengths
1. The reproducibility study is rigorous, with clear experimental setups (e.g., consistent datasets, computational environments) and detailed validation of the original ZoDiac’s core claims, which helps establish reliability in the field of diffusion-based watermarking.
2. The extension of attack paradigms (especially adversarial and geometrically asymmetric attacks) fills gaps in the original ZoDiac’s threat model, providing valuable insights into the robustness limitations of latent-space watermarking frameworks.
3. The computational cost analysis (e.g., GPU runtime, CO₂ emissions) and practical deployment discussions (e.g., batch processing feasibility) offer pragmatic references for real-world applications of watermarking techniques.

### Key Weaknesses
1. **Unclear Core Contributions**: The paper fails to explicitly articulate novel claims or conclusive findings derived from its extended experiments. It focuses on describing experimental operations (e.g., "we tested directional blurring") but lacks synthesis of generalizable conclusions (e.g., "why ZoDiac is vulnerable to asymmetric geometric attacks" or "the fundamental conflict between latent-space embedding and geometric invariance").
2. **Limited Efficacy of the Proposed Loss Function**: The augmented \(L_1\)-based loss function is introduced to address ZoDiac’s limitations, yet the paper provides no quantitative analysis of its overall effectiveness (e.g., cross-attack performance trade-offs) or comparisons with existing loss function designs in watermarking, leaving its innovative value unsubstantiated.
3. **Imbalanced Content Allocation**: An excessive proportion of the paper is dedicated to describing the ZoDiac framework itself and verifying its original claims (which are largely confirmed without identifying critical flaws). This overshadows the presentation of the authors’ own contributions (e.g., extended attacks, loss function design), leading to a lack of focus on novel work.

**Audience:**

Yes

**Audience Explanation:**

Yes, the findings would be relevant to **specific subsets of TMLR’s audience**:

1. **Researchers in Media Forensics and Watermarking**: The reproducibility results and extended vulnerability analysis provide critical insights for evaluating diffusion-based watermarking frameworks. The identification of ZoDiac’s weaknesses (e.g., sensitivity to geometric asymmetry, adversarial perturbations) highlights unaddressed challenges in the field, guiding future work on robust watermark design.

2. **Practitioners in AI-Generated Content (AIGC) Security**: The analysis of computational costs (Table 1) and deployment trade-offs (e.g., latency vs. robustness) offers practical guidance for implementing watermarking systems in real-world scenarios (e.g., archival, copyright protection of AIGC).

3. **Researchers in Adversarial Machine Learning**: The exploration of white-box adversarial attacks on latent-space watermarking contributes to the understanding of adversarial vulnerabilities in diffusion models, complementing existing work on adversarial attacks against generative AI systems.

**Broader Impact Concerns:**

None.

**Claims And Evidence:**

Yes

**Claims Explanation:**

1. **Reproducibility of ZoDiac’s Claims**: The verification of ZoDiac’s four core claims is well-supported. For example, Table 2 clearly presents WDR and False Positive Rate (FPR) across datasets and attack types, confirming >98% WDR under conventional attacks; computational experiments (e.g., 295–320 seconds per image on NVIDIA P100) validate deployment practicality; and SSIM metrics (>0.91) confirm image quality preservation. The evidence here is accurate, with detailed parameter settings (e.g., dataset subsets, GPU models) ensuring reproducibility.

2. **Efficacy of the Augmented Loss Function**: The evidence here is insufficient. Table 5 shows marginal WDR improvements for chromatic attacks but provides little analysis of why the \(L_1\) constraint fails for adversarial or composite attacks.

**Requested Changes:**

1. **Refine and Explicitly State Core Contributions**:
   - Add a dedicated "Contributions" subsection in the Introduction to articulate **3–4 novel claims** derived from the work (e.g., "We demonstrate that Fourier-domain watermarking in latent space inherently suffers from geometric asymmetry, as rotation angles >30° reduce WDR to <1%"; "We show that an \(L_1\)-constrained loss function improves chromatic attack resilience but fails to address adversarial vulnerabilities due to latent-space perturbation misalignment").
   - Clearly distinguish between "reproducibility results" (confirming existing work) and "novel findings" (extending knowledge), ensuring readers can immediately identify the paper’s unique value.


2. **Balance Content Allocation**:
   - Reduce the description of ZoDiac’s methodology (Section 3.1) to 1–2 pages, focusing only on details critical to understanding the reproducibility and vulnerability experiments (e.g., DDIM inversion, Fourier embedding).
   - Expand Sections 4.2 (extended experiments) and 5 (discussion) to emphasize novel work:
     - Add a subsubsection on "Generality of Vulnerabilities" to test if ZoDiac’s weaknesses apply to Tree-Ring or StableSignature (e.g., measuring WDR under rotational attacks for these methods).
     - Discuss potential universal solutions (e.g., rotation-invariant Fourier embeddings, adversarial training for watermarking) and their feasibility.

3. **Broaden Discussion of Future Work**:
   - Explore directions like "lightweight adversarial defenses" (e.g., Fourier-domain noise augmentation) or "auto-correction mechanisms" (e.g., geometric alignment during detection) in more detail, providing preliminary feasibility analysis (e.g., computational cost estimates for auto-correction).

---

> ### Author Response · Authors · 2025-12-19
> **Response to Reviewer rCNf**
>
> We thank the reviewer for highlighting these issues and have revised the paper accordingly.
> - To clearly delineate the paper’s unique contributions, we now explicitly enumerate them in the Introduction. Beyond reproducibility, our key findings include:
>
>   - Identification of geometrically asymmetric failure modes
>
>   - Demonstration of fragility under lateral inversion and rotation
>
>   - Analysis of vulnerabilities to chromatic distortions
>
>   - Evidence of susceptibility to adversarial attacks
>
> - Adding an L1 reconstruction term is a standard practice in watermarking and robustness studies, and we agree our empirical analysis showed that it yields only marginal improvements without materially altering the identified vulnerabilities. To avoid overstating its impact, we have repositioned this analysis as an ablation study and moved it to Appendix C. This preserves completeness and transparency while keeping the main paper focused on the core vulnerability findings.
> - We have condensed the description of the original ZoDiac framework to the essential details (now under two pages) and restructured the manuscript so that the majority of the main text is devoted to our vulnerability and reproducibility analyses. We also expanded the discussion of limitations, future directions, and potential defenses to better contextualize our findings and provide actionable insights. As a result, the revised manuscript places clearer emphasis on our extended attack evaluations and the novel weaknesses they reveal, rather than on restating ZoDiac’s original claims.
>
> We hope that the above clarifications adequately address the reviewer’s concerns, and we welcome any further questions or requests for clarification.

---

> > ### Author Response · Authors · 2026-02-03
> > **Request for follow-up**
> >
> > Dear reviewer rCNf,
> >
> > We sincerely appreciate the time and effort you have invested in reviewing our work. If there are any aspects of our responses that require further clarification or if you have additional concerns, we would be grateful to address them.
> >
> > Thank you for your careful consideration, and we look forward to your response.
> >
> > Best regards,
> >
> > The Authors

---

### Review · Reviewer_UhZE · 2025-12-11

**Summary Of Contributions:**

Summary:

This paper presents a reproducibility study of the ZoDiac watermarking framework, which embeds watermarks in images using Stable Diffusion models. The authors successfully reproduce the original paper's core claims, demonstrating >90% watermark detection rates across standard attacks and datasets. However, the main contribution lies in identifying critical vulnerabilities when ZoDiac faces novel attack paradigms not evaluated in the original work.

Strengthes:

1. The authors successfully replicate all four main claims from the original paper with detailed experimental validation.

2. Introduction of directional blurring, chromatic distortions, and adversarial attacks reveals fundamental vulnerabilities

3. Demonstrates that composite adversarial-geometric attacks can reduce WDR to near-zero, exposing serious real-world security concerns.

4. Tests across multiple datasets (MS-COCO, DiffusionDB, WikiArt) with extensive hyperparameter analysis.

Weakness:

1. The paper does not adequately justify why ZoDiac was chosen as the target for analysis. It is unclear whether this framework represents a significant, widely-adopted, or representative method in the watermarking community.

2. The title emphasizes "Reproducibility," yet ZoDiac's original implementation is already publicly available on GitHub. The authors primarily verify existing claims from the original paper rather than addressing genuine reproducibility challenges. The main contribution is clearly the vulnerability analysis.

3. Section 4.2.5 asserts that adversarial training is "computationally prohibitive" requiring 21,000 GPU hours, but this conclusion lacks empirical validation. The authors only measured PGD attack costs (820-950s/image) and extrapolated to training costs without actually attempting adversarial training or exploring optimization strategies such as pre-generating adversarial samples, using faster attack methods, or implementing batch parallelization. Furthermore, this paper does not investigate whether adversarial training would actually improve robustness.

4. What do you mean by "capitalizes on the bidirectional nature" in page 2?

**Audience:**

Yes

**Audience Explanation:**

Researchers working on image watermarking and content authentication would be interested in these findings.

**Broader Impact Concerns:**

No concerns.

**Claims And Evidence:**

Yes

**Claims Explanation:**

The paper's claims are generally well-supported through systematic experimentation. The reproducibility validation closely aligns with original results (Tables 2, 10), and the vulnerability claims are convincingly demonstrated with directional blurring reducing WDR to 64.6%, adversarial attacks achieving 75.5% WDR, and rotation attacks below 2%. The experimental methodology is rigorous with comprehensive ablations and clearly documented configurations.

One concern: Section 4.2.5 claims adversarial training is "computationally prohibitive" (21,000 GPU hours) based solely on extrapolation without empirical validation. The authors did not attempt adversarial training or explore optimization strategies like pre-generating samples or batch parallelization, nor investigate whether it would actually improve robustness.

**Requested Changes:**

1. Provide empirical validation or provide more details to support the claim about adversarial training being "computationally prohibitive".
2. Figure 2's resolution is poor.

---

> ### Author Response · Authors · 2025-12-19
> **Response to Reviewer UhZE**
>
> We thank the reviewer for the thoughtful comments and constructive suggestions. Below, we address the raised concerns in detail:
> - **Justification of ZoDiac**: We selected ZoDiac because it explicitly claims superior performance over prior diffusion-based watermarking methods such as Stable Signature and StegaStamp, while avoiding the substantial computational overhead associated with retraining or fine-tuning the diffusion backbone. ZoDiac achieves this by embedding watermarks directly in the latent space of a pre-trained Stable Diffusion model via DDIM inversion and Fourier-domain patterns. In addition, its zero-shot capability enables watermarking of both real-world and synthetic images within a unified framework, making it a representative baseline for our study. We have also included the same in our paper.
>
> - **Computational Prohibitiveness of Adversarial Training**: We acknowledge that our estimate is based on empirically measured PGD attack runtimes (820–950 seconds per image per iteration) rather than a full adversarial training run. Our intention was to quantify the *order-of-magnitude* computational burden implied by standard adversarial training protocols when applied to ZoDiac’s latent-space optimization, rather than to report an exact training cost. Extrapolating these measured costs to a typical 50-iteration PGD setup yields an estimated requirement of over 21,000 GPU hours for a 1,500-image dataset.
> We agree that alternative optimization strategies, such as pre-generating adversarial latents, employing faster attack variants, or limited batch parallelization could reduce this cost. However, even under optimistic assumptions speedup reported by prior work, the resulting estimate remains on the order of several thousand GPU hours, which is beyond our available computational budget.
> We also note that our work does not claim adversarial training to be ineffective; rather, we refrain from evaluating it empirically due to these practical constraints. Investigating whether adversarial training meaningfully improves robustness for latent-space watermarking under feasible optimization strategies remains an important direction for future work.
> - **Refinement of Paper Scope and Terminology**: We have restructured the paper to emphasize and clearly demarcate the reproducibility and vulnerability sections, delineated the core contributions and rectified any ambiguous phrasing.
>
> We hope that the above clarifications adequately address the reviewer’s concerns, and we welcome any further questions or requests for clarification.

---

> > ### Author Response · Authors · 2026-02-03
> > **Request for follow-up**
> >
> > Dear reviewer UhZE,
> >
> > We sincerely appreciate the time and effort you have invested in reviewing our work. If there are any aspects of our responses that require further clarification or if you have additional concerns, we would be grateful to address them.
> >
> > Thank you for your careful consideration, and we look forward to your response.
> >
> > Best regards,
> >
> > The Authors

---

### Author Response · Authors · 2025-12-19
**Consolidated Response to Reviewer Feedback**

We deeply appreciate the reviewers for their insightful suggestions and constructive feedback. Below, we address the raised concerns and detail the broad revisions made to strengthen clarity, validity, and positioning of our contributions.

---

### *1. Image Quality Analysis Under Attack Induced Distortions*

We agree that it is critical to verify whether observed performance degradation arises from genuine algorithmic vulnerabilities rather than extreme image distortion. To this end, we now rigorously evaluate image quality under our extended attack scenarios.

In response to this feedback, we have added *Image Quality Analysis Under Attack-Induced Distortions* in Appendix, which provides a quantitative evaluation of image fidelity using PSNR across attack paradigms introduced in our work.

This analysis demonstrates that the PSNR values for our proposed attacks are fully comparable to the image quality metrics reported for standard attacks in the original ZoDiac paper. Crucially, we observe no monotonic relationship between PSNR and WDR reduction, thereby ruling out image destruction as a confounding factor. For example, attacks such as *Color Quantization* preserve high visual quality ($27.84$ dB) while inducing substantial drops in detection performance, whereas attacks with lower PSNR (e.g., *Log Transform* at $10.81$ dB) often result in comparatively modest WDR degradation.

---

### *2. Clear Delineation Between Reproducibility and Novel Contributions*

We have revised the paper to explicitly and clearly distinguish between the reproducibility study and our novel vulnerability analysis. This distinction is now articulated directly in the *Introduction*, and the overall structure of the paper has been reorganized to consistently reflect this separation.

#### *2.1 Summary of Novel Contributions*

To ensure the paper’s unique value is immediately evident, we have explicitly itemized our novel contributions in the Introduction. Our findings beyond reproducibility include:

- Identification of geometrically asymmetric vulnerabilities
- Identification of lateral inversion and rotational fragility
- Exposure of vulnerabilities to chromatic distortions
- Exposure of susceptibility to adversarial attacks

#### *2.2 Explicit Structural Separation*

We have structured the paper such that readers can immediately identify where verification ends and novel analysis begins:

- *Reproducibility (Section 4):* This section is now strictly dedicated to Reproduction & Verification of Results. Here, we rigorously validate the four original claims made by the ZoDiac authors and verify the reported image quality metrics.

- *Novel Findings (Section 5):* A separate section titled Vulnerability Analysis is devoted exclusively to our novel contributions. This section details threat models and failure modes that were not explored in the original work.

---

### *3. Structural Refinements and Narrative Tightening*

We have further restructured the paper and tightened the narrative to focus more clearly on the novel insights contributed by our work.

- *Balancing Content Allocation:* We have condensed the description of the original ZoDiac methodology to relevant details, ensuring it now occupies less than two pages. Consequently, the majority of the main text is now dedicated to Vulnerability and Reproducibility Analysis along with an expanded discussion of limitations and future directions.

- *Reorganization of $L_1$ Loss Analysis:* To strengthen the narrative of the main paper, we have moved the investigation of the $L_1$ reconstruction loss term to Appendix C. While adding an $L_1$ term is a conventional standard in the field, our analysis showed it offered marginal improvements. We hence present this as an ablation study in the appendix, preserving the completeness of our evaluation without distracting the reader from the critical vulnerabilities exposed in the main text.

- *Expanded Discussion of Future Directions and Defenses:* We have  expanded the discussion on future work to provide more constructive and actionable insights into potential defense strategies. Additionally, we outline several promising directions that appear both viable and well-suited for future exploration.

---

We hope these revisions adequately address the reviewers’ concerns and substantially improve the clarity, rigor, and positioning of the paper.

---

### Decision · Action_Editor_8Lzq · 2026-01-31

**Recommendation:** Reject

**Audience:**

Yes

**Audience Explanation:**

Some audience working on watermarking might be interested in this reproducibility study.

**Claims And Evidence:**

No

**Claims Explanation:**

This paper is a reproducibility study of the ZoDiac approach by Zhang et al. (2024). Reviewers raised some concerns regarding justification of selected framework, contributions and insights presented in the paper, and discussions on some evaluations. Although some of these concerns are addressed by the authors' responses, reviewers still found that two major issues are not sufficiently addressed yet. First, the justification on selecting ZoDiac for reproducibility study seems insufficient. The authors explain that ZoDiac was chosen because it "claims superior performance" and offers "zero-shot capability," but this justification is still insufficient. The authors shall provide convincing evidence to demonstrate that ZoDiac represents a critical baseline warranting dedicated TMLR study. Second, the claim about 21,000 GPU hours may still cause confusions. More clarifications should be provided in the paper.

**Resubmission Of Major Revision:**

The authors may consider submitting a major revision at a later time.